# OpenReview forum: "Robust RLHF with Noisy Rewards"
_ICLR.cc/2025/Conference — ICLR 2025 Conference Withdrawn Submission_

### Official Review · Reviewer_igjG · 2024-10-16

**Soundness:** 2
**Presentation:** 2
**Contribution:** 2
**Rating:** 3
**Confidence:** 4

**Summary:**

The paper proposed a contrastive reward method that intends to address possible noisy rewards from inaccurate reward models. The authors justify the effectiveness of their methods by empirical results on models alignment.

**Strengths:**

The paper paid attention to a direction of the existing alignment research which is less explored, however quite natural: the reward model can be (actually indeed) inaccurate. The authors thus proposed subtracting the baseline reward function from usual reward utilized in PPO algorithm for RLHF.

**Weaknesses:**

**Writing**: The overall writing has a significant space to improve. There are confusing descriptions and notations, making some parts of the paper hard to follow and understand. For example, Section 2.1, which should be the most important section to introduce the notion of Robust RLHF, is very short and not elaborated enough to clearly explain the motivations and formulation details of this paper. In addition:
* Table 1 is rather confusing. Is this error rate the dataset’s error rate or the error rate of the reward model?
* In Equation (4) and (5), there are many new pop-up elements: what is the second expectation for in (4) and (5)? I think in (4) the second expectation should be $\Psi$ right? What is $\mathcal{D}_{RL}$ in (5)? Also, are there any reasons why authors do not consider any type of regularization, like KL regularization in RLHF, but include this regularization back in the practical algorithm design?
* On line 123-125: The confusion function needs to be formally defined. The results for analysis are also limited to the discrete-value setting which seems to be rather restricted and undermine the generalizability of the theorem results.
* Robustness in 129-131: Here the definition of such robustness is unclear and confusing to me. Since the paper is developing around robustness, a readily defined definition shall be necessary.

**Novelty**: I am also confused about the novelty of this paper, as the proposed mechanism for increasing robustness and its relation to resulting algorithms is still unclear after my reading. The final proposed improvement over PPO baseline looks to be subtracting a baseline function from reward, and this is a standard practice even in RLHF original paper, like normalization to mean 0 for reward models. It is also a widely adopted technique for variance reduction, as the author also mentioned on line 196-198. It will be beneficial to further explain the version and details of PPO the paper utilized for baseline comparison. Any further evidence of noisy reward and how algorithms behave with respect to the noisy level of the reward models shall be a better way to showcase the effectiveness.

**Possible missing references**: are there any connections between the approach in this paper and other papers like MaxMinRLHF[1] or other robust preference optimization papers? There lacks a discussion of existing literature on this line.

[1] MaxMinRLHF: https://arxiv.org/pdf/2402.08925

**Questions:**

* Please see questions in weakness
* Have the authors tried aligning models with proposed algorithms and test the resulting models’ performance on benchmarks like AlpacaEval?

---

> ### Author Response · Authors · 2024-11-25
>
> Thank you for all your valuable questions and comments. Below, we provide detailed responses and clarifications to address the points raised.
> ### **Theoretical Supplement**
>
> #### **Clarifying Table 1 and Equation [4]**
>
> Table 1 shows the reward model's (RM) high error rate, indicating significant noise affecting its accuracy. Addressing this noise during RL training is crucial for improving policy performance. We analyze the theoretical impact of reward noise and propose robust mitigation methods.
>
> **Error in Equation [4]**: The second expectation should indeed be $\(\Psi\)$.
>
> $\(D_{RL}\)$ represents the set of prompts used during RL, where prompts $(\(x\))$ query the actor model to generate responses $(\(y\))$. While our practical algorithm includes KL regularization, $\(\Psi\)$'s simplified formulation highlights its role in improving RM robustness. Our focus is on $\(\Psi\)$'s theoretical properties, addressing reward noise issues.
>
> ---
>
> #### **Formal Definition of the Confusion Matrix**
>
> The confusion matrix \(C\) is defined as:
>
> $$
> C =
> \begin{bmatrix}
> c_{1,1} & c_{1,2} & \cdots & c_{1,K} \\
> c_{2,1} & c_{2,2} & \cdots & c_{2,K} \\
> \vdots & \vdots & \ddots & \vdots \\
> c_{K,1} & c_{K,2} & \cdots & c_{K,K}
> \end{bmatrix},
> $$
>
> where \(K\) is the number of reward levels, and each element \(c_{i,j}\) is:
>
> $$
> c_{i,j} = \mathbb{P}(r_{\psi} = j \mid r^* = i),
> $$
>
> the probability of the noisy reward model \(r_{\psi}\) predicting \(j\) when the true reward is \(i\).
>
> For the binary case (\(K = 2\)):
>
> $$
> C =
> \begin{bmatrix}
> c_{0,0} & c_{0,1} \\
> c_{1,0} & c_{1,1}
> \end{bmatrix},
> $$
>
> where $\(c_{0,1}\)$ (error when $\(r^* = 0\)$ but $\(r_{\psi} = 1\)$) is $\(c_0\)$, and $\(c_{1,0}\)$ (error when $\(r^* = 1\)$ but $\(r_{\psi} = 0\)$) is $\(c_1\)$ in the paper.
>
> The original manuscript omitted the confusion matrix to reduce notation complexity. However, we now include it for completeness.
>
> ---
>
> #### **Limitation to Discrete Random Variables**
>
> Our analysis focuses on discrete random variables, limiting its generalizability to continuous settings. To address this:
>
> 1. Discretize continuous variables into fine-grained levels for approximation.
> 2. Increase granularity to converge to the continuous case.
>
> This provides a practical, though not fully rigorous, framework for continuous variables.
>
> ---
>
> #### **Definition of Robustness**
>
> **Robustness** is defined as:
>
> $$
> \pi^*_{r_{\psi}}(\Psi) \rightarrow \pi^*_{r^*}(\Psi),
> $$
>
> where $\(\pi^*_{r_{\psi}}(\Psi)\)$ is the optimal solution using the noisy reward model $\(r_{\psi}\)$, and $\(\pi^*_{r^*}(\Psi)\)$ is the optimal solution with the true reward model $\(r^*\)$. Robustness ensures that noise in $\(r_{\psi}\)$ does not affect optimization results. This definition will be added to the revised manuscript for clarity.
>
> ---
>
> ### **Addressing Novelty Concerns**
>
> Our work offers a novel approach to mitigating reward noise in RL:
>
> 1. **Distinct Method**:
>    Rather than subtracting baselines or calibrating offline rewards, we derive robustness properties under noise theoretically.
>
> 2. **Theoretical Contributions**:
>    - **Affine Transformation Robustness (Theorem 1)**: Demonstrates robustness to reward noise.
>    - **Contrastive Penalty Reward (Theorem 2)**: Proposes a framework with theoretical advantages in noise reduction.
>
> 3. **Experimental Results**:
>    Besides main experiement, the synthetic experiments also show significant performance improvement under high noise, providing actionable insights for RLHF practitioners.
>
> ---
>
> ### **On Missing References**
>
> We acknowledge missing the MaxMin-RLHF paper, which uses EM algorithms to address reward mismatch, closely related to our work. This will be cited in the revised manuscript. Other relevant works are already discussed in the "Related Work" section.
>
> ---
>
> ### **AlpacaEval Results**
>
> Our **Llama3.1-ours** model achieved a 23.9% LC Win Rate on AlpacaEval, surpassing **Mixtral 8x7B v0.1** (leaderboard) and **Llama3.1-instruct** (20.9%). These results validate our method's real-world effectiveness.

---

> > ### Author Response · Authors · 2024-12-02
> >
> > Dear Reviewer,
> >
> > I hope this message finds you well. We sincerely appreciate the valuable feedback you provided on our paper. As the discussion period will soon come to a close, we kindly ask if you could take a moment to review our responses to your comments. Your insights are incredibly important to us and will help ensure we address any remaining concerns.
> >
> > If our responses meet your expectations, we would be grateful if you could consider adjusting your rating accordingly.
> >
> > Thank you very much for your time and effort. We truly value your contribution to our work.
> >
> > Warm regards,
> >
> > The Authors

---

> > > ### Comment · Reviewer_igjG · 2024-12-03
> > >
> > > I would like to the thank the authors for responses and extra efforts on experimental results. Despite this, my concern that the connection between theorems proved and algorithm yielded is rather vague is not addressed, and I also have to say that the novelty of the resulting algorithms beyond standard variance reduction techniques for RL/RLHF is still rather limited. Nevertheless, I would like to thank the authors for their efforts on addressing my concern.

---

### Official Review · Reviewer_xGEj · 2024-10-30

**Soundness:** 4
**Presentation:** 4
**Contribution:** 2
**Rating:** 6
**Confidence:** 3

**Summary:**

The authors address the noise in the reward model by introducing a contrastive reward function that compares the reward of a generated response to the average reward of baseline responses generated by a previously trained model. They show this simple modification outperforms the vanilla PPO algorithm by 20%.

**Strengths:**

The paper is well-written and easy to follow. The experimental methodology is rigorous, with many ablation studies, and the authors even employed human annotators. The method is intuitive and easy to implement.

**Weaknesses:**

The idea of subtracting the value of a reference answer to reduce reward model variance between various prompts [has been there at least since January 2023.](https://wandb.ai/carperai/summarize_RLHF/reports/Implementing-RLHF-Learning-to-Summarize-with-trlX--VmlldzozMzAwODM2#gotcha-1:-normalization) I am not aware of any research papers on this topic, but it makes me question the impact of this paper if the practitioners already use this method.

**Questions:**

I like the paper, and I think it is quite complete; I am only a bit sceptical about the novelty of this work, hence a lower score. As pointed out by the authors, it would be interesting to see how well the method performs over multiple rounds, recalibrating the base rewards after each iteration.

---

> ### Author Response · Authors · 2024-11-25
>
> ### Key Differences and Contributions
>
> #### **1. Variance Reduction Through Aggregation**
>
> - The algorithm in the linked paper significantly differs from ours in how it handles reference answers:
>
>   - Subtracts **only one reference answer**, leading to high variance due to fluctuations in individual responses.
>   - **Our Method** Aggregates **multiple responses**, reducing variance and yielding more stable and consistent rewards.
>
> - **Empirical Evidence**:
>   To highlight this difference, we conducted an experiment using our **in-house test set** with **GPT-4 Turbo as the judge**. The results demonstrate the superiority of our approach over the baseline:
>
>   | **Comparison**                         | **Win Rate**        |
>   | -------------------------------------- | ------------------- |
>   | **Single Response vs. Average Reward** | **37% : 10% : 53%** |
>
>   - **Interpretation**: Subtracting only one reference answer introduces instability, often resulting in overly high or low rewards. In contrast, our aggregation method ensures more robust and consistent performance.
>
> ---
>
> #### **2. Theoretical Justification**
>
> - The linked algorithm is presented **empirically**, but it lacks both a **theoretical foundation** and a **clear underlying mechanism**.
>
> - **Our Contributions**:
>   We provide the **first theoretical analysis** in this domain, distinguishing our method from the baseline:
>   - **Affine Transformation Robustness (Theorem 1)**:
>     We derive our formula through an affine transformation of the true preference, inherently enhancing robustness to noise.
>   - **Contrastive Penalty Reward (Theorem 2)**:
>     We establish the theoretical advantages of our contrastive penalty reward, further validating its effectiveness.
>
> ---
>
> ### Insights on Iterative Recalibration of Rewards
>
> - **Experimentation and Challenges**:
>   We tested iterative recalibration of the reward, as suggested, but encountered the following issues:
>   - **Reward Hacking**: The model began exploiting the entire valid reward range, leading to extreme outputs with excessively high rewards.
>   - **Prolonged Training Effects**: Extended training exacerbated this issue, producing unusual outputs similar to challenges observed in vanilla PPO.
>
> - **Mitigation Strategies**:
>   To address reward hacking, we implemented **early stopping**, which:
>   - Delayed the onset of reward hacking.
>   - Improved overall performance compared to vanilla PPO.
>
> - While early stopping proved effective, further experimentation is required to fully quantify its impact. Unfortunately, due to computational resource constraints, we are unable to provide conclusive results at this time.
>
> We appreciate your valuable suggestion regarding iterative recalibration and will explore this direction further in future work. Our findings, both theoretical and empirical, aim to contribute meaningfully to advancements in this area. Thank you for your insights, which have been instrumental in driving this research forward.

---

> > ### Comment · Reviewer_xGEj · 2024-11-27
> >
> > Thank you for providing a thorough response. Upon reading other reviews and getting familiar with the related work other reviewers listed, I believe a common consensus questioning the novelty of this work is in place. I am keeping my score.

---

### Official Review · Reviewer_tkww · 2024-11-03

**Soundness:** 3
**Presentation:** 3
**Contribution:** 1
**Rating:** 1
**Confidence:** 4

**Summary:**

The paper proposes to use offline generated rewards to calibrate the reward at training time in the RLHF pipeline.

**Strengths:**

The paper rediscovers the use of a baseline in policy gradient methods.

**Weaknesses:**

This method is similar to RLOO or VinePPO in the one turn case. The only slight difference is that the calibrated reward is produced offline.
However, it seems that this offline calibration has already been published in this study:
https://arxiv.org/pdf/2410.01679 (see section 4.1.1 called Calibrated Regularized Policy Gradient)
I think the work has been produced concurrently and therefore there is no added scientific contribution.

**Questions:**

How different is your method from existing methods that subtract the average reward to the reward?

---

> ### Author Response · Authors · 2024-11-25
>
> ### Key Differences Between Our Work and VinePPO
>
> #### **1. Theoretical Perspective**
>
> - **Universal Framework for Reward Robustness**:
>   Our work introduces a **universal framework** to improve reward robustness during reward model (RM) inference. This framework is grounded in strong theoretical foundations:
>   - **Affine Transformation of True Preference**:
>     The contrastive reward is rigorously derived through an affine transformation of the true preference, inherently ensuring robustness to noise (*Theorem 1*).
>   - **Contrastive Penalty Reward**:
>     We establish significant theoretical advantages for the proposed contrastive penalty reward, demonstrating its **effectiveness and reliability** in practice (*Theorem 2*).
>
> - **Lack of Theoretical Foundation in VinePPO**:
>   VinePPO does not provide a comparable theoretical understanding or a framework to ensure reward robustness.
>
> ---
>
> #### **2. Empirical Comparison**
>
> - **Performance in Math Tasks**:
>   - VinePPO struggles to compete with value function-free methods like **RLOO** and **GRPO**.
>   - Our method demonstrates **consistent advantages** across general tasks, including code-related ones.
>
> - **Fundamental Motivations**:
>   - VinePPO relies on **Monte Carlo (MC)-based estimates** to bypass the value function.
>   - Our method directly addresses the challenges of **imperfect reward models**, leading to **improved robustness** in the Proximal Policy Optimization (PPO) algorithm.
>
> - **Efficiency and Latency**:
>   - Unlike VinePPO, which requires **sampling nine trajectories**—an impractical budget for training large language models—our approach is computationally **more efficient**.
>   - By pre-collecting baseline responses, we reduce latency and consistently improve performance across all prompt dimensions.
>
> - **Value Function-Free vs. Value Function-Based Methods**:
>   - Under the same computational budget, VinePPO and other value function-free methods fail to outperform methods that use value functions.
>
> ---
>
> #### **3. Experimental Results**
>
> - **Higher Win Rate Against VinePPO**:
>   We implemented VinePPO for a **fair comparison**. Under the same sampling budget (5) and after 500 steps, our method demonstrates a **higher win rate**:
>
>   | **Evaluation Metric** | **Ours** vs. **VinePPO** |
>   | --------------------- | ------------------------ |
>   | **GPT-4-as-a-judge**  | **46.4% : 22.6% : 31%**  |
>
> ---
>
> #### **4. Reward Normalization Comparisons**
>
> - **Baseline Comparisons**:
>   We compared our approach against static normalization and hard clipping methods that subtract the average reward.
>
> - **Strength of Our Method**:
>   Our **baseline reward** and **dynamic reward scaling** methods significantly outperform vanilla reward normalization approaches:
>
>   | **Evaluation Metric** | **Ours** vs. **Reward Normalization** |
>   | --------------------- | ------------------------------------- |
>   | **GPT-4-as-a-judge**  | **36.1% : 34.2% : 29.7%**             |

---

> > ### Comment · Reviewer_tkww · 2024-11-26
> > **Rebuttal feedback**
> >
> > I think that there is a lack of novelty in this work. For this reason, I will not change my score and advocate for rejection.

---

### Official Review · Reviewer_JLkV · 2024-11-04

**Soundness:** 3
**Presentation:** 3
**Contribution:** 3
**Rating:** 8
**Confidence:** 3

**Summary:**

This paper, titled "Robust RLHF with Noisy Rewards," addresses a critical challenge in reinforcement learning from human feedback (RLHF): the vulnerability of reward models to noise. The authors propose a novel approach using contrastive rewards to enhance RLHF robustness by penalizing uncertainty, promoting stability, and improving learning focus. The paper includes theoretical justifications, as well as extensive empirical results that demonstrate the effectiveness of the proposed method, showing consistent improvements over established baselines. This paper’s contribution could have a meaningful impact on real-world RLHF applications where noisy feedback is prevalent.

**Strengths:**

1. Clear motivation: The paper emphasizes a common issue (noisy reward signals) in RLHF and offers a practical solution.
2. Theoretical backing: The authors justify the contrastive reward mechanism with robust mathematical grounding.
3. Empirical support: Extensive experiments with both human and automated evaluation show meaningful performance gains.

**Weaknesses:**

1. Limited baseline comparisons: The paper benchmarks against some strong baselines but lacks comparisons to more recent methods that handle reward noise differently.
2. Applicability to real-world scenarios: The method assumes a binary reward noise model, which may oversimplify complex noise distributions in actual data.

**Questions:**

1. Could the approach be combined with other regularization techniques to further enhance stability?
2. Are there plans to address other types of noise beyond reward model noise to make the framework more comprehensive?

---

> ### Author Response · Authors · 2024-11-25
>
> #### **1. Limited Baseline Comparisons**
>
> We appreciate the reviewer’s observation regarding the limited baseline comparisons. Our method fundamentally differs from previous approaches like **Offset [1]**, which aim to reduce reward noise **before training** by minimizing reward bias in the dataset.
>
> - **Key Difference**:
>   Unlike Offset and similar methods, our approach focuses on **optimizing the LLM model** using the noisy reward model **during RL training**. This distinction highlights the unique nature of our method and its contribution to improving robustness in reinforcement learning with imperfect reward signals.
>
> ---
>
> #### **2. Oversimplification of Complex Noise Distributions**
>
> We thank the reviewer for their perceptive insight into the complexity of noise distributions in real-world data.
>
> - **Current Work**:
>   We provide a **brief analysis** for multi-level reward settings in **Appendix C**, demonstrating that our framework can generalize to multiple reward models, provided the reward signals can be discretized.
>
> - **Future Plans**:
>   We recognize that further exploration of complex noise distributions is a promising direction and are excited to investigate this topic in greater depth in future work.
>
> ---
>
> #### **3. Plans to Address Other Types of Noise**
>
> We sincerely thank the reviewer for bringing up this intriguing direction.
>
> - **Future Work**:
>   We find this suggestion highly interesting and are eager to extend our framework to address other types of noise in future studies. We believe this line of work will further enhance the robustness and applicability of our method in diverse real-world scenarios.
>
> ---
>
> ### Reference
>
> [1] Park, Junsoo, et al. "Offsetbias: Leveraging debiased data for tuning evaluators." *arXiv preprint arXiv:2407.06551* (2024).

---

### Note · Authors · 2024-12-27

I have read and agree with the venue's withdrawal policy on behalf of myself and my co-authors.